# Regional Conservation and Transcriptional Regulation of Tumor-Associated Genes by macroH2A1 Deposition in Mammalian Cells

**DOI:** 10.3390/biom15101386

**Published:** 2025-09-29

**Authors:** Yongzhuo Deng, Zeqian Xu, Le Zhang, Bishan Ye, Zhifeng Shao, Xinhui Li

**Affiliations:** School of Biomedical Engineering, Shanghai Jiao Tong University, Shanghai 200240, China; yzdeng@sjtu.edu.cn (Y.D.); xzq_1995@sjtu.edu.cn (Z.X.); zlehit@sjtu.edu.cn (L.Z.); yip_33@sjtu.edu.cn (B.Y.); zfshao@sjtu.edu.cn (Z.S.)

**Keywords:** macroH2A1, mitosis, nChIP seq, transcriptional regulation, tumorigenesis

## Abstract

Histone variant macroH2A1 (mH2A1) has been widely recognized as a suppressor of gene expression. Recently, a cell cycle-dependent deposition of mH2A1 was discovered in mouse cells, but whether this process exists in human chromatin is unclear, which might be crucial for related diseases, particularly cancer. In this study, with native chromatin immunoprecipitation (nChIP-seq), we firstly demonstrate that dynamic mH2A1 domains occur in both normal and cancerous human cells and have conserved enrichment patterns across species. Our findings further provide new epigenetic insights into the role of mH2A1 in malignant proliferation, offering a novel perspective for future cancer research.

## 1. Introduction

As the fundamental units of eukaryotic chromatin, nucleosomes are composed of an octamer containing four core histones-H2A, H2B, H3, and H4 and their variants [1,2]. Among these variants, macroH2A1 (mH2A1) stands out due to its unique structural and functional characteristics [3]. Structurally, mH2A1 is approximately three times larger than canonical H2A, owing to a distinctive 30 kDa mdomain at its C-terminus [4]. Functionally, mH2A1 plays a vital role in transcriptional repression and X chromosome inactivation [5,6]. Genome-wide, mH2A1-containing nucleosomes form broad domains that frequently overlap with heterochromatin regions [7,8]. During mitosis, these domains are reassembled de novo on highly expressed genes, while their interphase distribution remains largely stable [9]. This dynamic behavior underscores mH2A1’s importance in regulating both the transcriptome and cell cycle progression.

Over the past decades, histone variants have been implicated in the transcriptional regulation of key genes in cancer cells [10]. Among them, mH2A1 is particularly notable due to its regulatory functions, suggesting a potential role in tumorigenesis. As a transcriptional repressor, mH2A1acts as a tumor suppressor by maintaining genomic stability and silencing oncogenic pathways [11]. However, the aberrance of its isoforms mH2A1.1 and/or mH2A1.2 is frequently observed in malignant tumors and is associated with poor prognosis and increased malignancy [11,12,13]. Paradoxically, emerging evidence also indicates that mH2A1 can exhibit oncogenic properties in certain contexts. For example, its elevated expression promotes tumor progression in non-small cell lung cancer and facilitates an epithelial-to-mesenchymal transition (EMT) phenotype in breast cancer [14]. These contrasting roles underscore the complexity of mH2A1 function in cancer and highlight the importance of understanding its context-dependent mechanisms. Such insights are critical for designing targeted epigenetic therapies that modulate mH2A1 activity.

To investigate these questions, we perform native chromatin immunoprecipitation (nChIP-seq) followed by sequencing in G1/S phase and mitotically synchronized human cells (ARPE-19 and HeLa). Our results confirm that mH2A1 is reloaded onto chromosomes prior to mitosis and removed afterward [9], indicating that mH2A1 deposition is a conserved mechanism across species. Notably, the shared protein-coding genes marked by dynamic mH2A1 domains are primarily involved in essential cellular functions. In HeLa cells, we further identify a cancer-specific pattern of dynamic mH2A1 distribution, which is associated with a pro-oncogenic gene. Together, these findings reveal a conserved mechanism of mH2A1 dynamics during the cell cycle and highlight its potential as an epigenetic marker for investigating cancer-associated gene regulation.

## 2. Materials and Methods

### 2.1. Cell Culture and Cell Cycle Synchronization

All cell lines were kept in a humidified incubator at 37 °C temperature with a 5% CO_2_ supply and cultured in DMEM (GIBCO, Carlsbad, CA, USA) with 10% FBS (GIBCO, Carlsbad, CA, USA) and 1% Pen/Strep anti-biotin (GIBCO, Carlsbad, CA, USA). For G1/S synchronization, we cultured cells under starvation conditions (DMEM with 0.5% FBS and 1% Pen/Strep) for 48 h, followed by the addition of fresh medium containing 1% Aphidicolin (Abcam, Cambridge, MA, USA) for 18 h before collection. To obtain mitotic cells, cells were cultured to reach approximately 80% confluency then treated with 100 ng/mL colcemid (Sigma-Aldrich, St. Louis, MO, USA) for 12 h, after which mitotic cells were shaken off and collected for purifying the mitotic chromosome. Synchronization efficiency was assessed using PI staining with flow cytometry, and we also performed Western blotting (WB) to semi-quantitatively assess the mH2A1 enrichment and validate antibody specificity (Appendix A).

### 2.2. Nucleosome Preparation and Nchip-Seq

The preparation for nucleosomes was optimized from previously described procedures [9,15,16]. For synchronized mitotic cells, we removed culture medium and washed with phosphate-buffered saline (PBS). Mitotic cells were shaken off by mechanical tapping and collected by centrifuging at 120× *g* for 5 min. The cell pellet was resuspended in hypotonic buffer for 30 min lysis at room temperature. The membrane-penetrated cells were centrifuged at 600× *g* for 5 min at 4 °C and resuspended in PA buffer to preserve chromosomal integrity. Cell resuspension was homogenized with a 7 mL Dounce homogenizer for 10 min with intermittent pauses. The homogenate was centrifuged at 190× *g* for 5 min at 4 °C to remove large debris. The supernatant was filtered through 10 μm and 5 μm filters sequentially. Chromosomes were centrifuged at 1750× *g* for 10 min at 4 °C and resuspended in MNase (micrococcal nuclease) buffer (15 mM Tris-HCl pH 7.5, 0.5 mM EGTA,2 mM EDTA, 80 mM KCl, 20 mM NaCl, 0.5 mM spermidine, 0.2 mM spermine, 35 mM CaCl_2_, 35 mM Tris-HCl pH 8.0, 10 μg/mL BSA, 1 mM PMSF, and 1 mM protease inhibitor cocktail); mitotic chromosomes were digested with MNase (3000 gel units/mL, NEB, Ipswich, MA, USA) at 4 °C overnight, and the digestion was terminated by 20 mM EDTA. Mitotic nucleosomes were centrifuged at 10,000× *g* at 4 °C.

G1/S synchronized cells were collected by trypsin digestion at 37 °C for 2 min, then washed twice with PBS before being suspended in MNase buffer containing 0.5% NP-40 to release chromatin. For G1/S chromatin digestion, MNase was performed with 2000 gel units/mL under equivalent conditions at 4 °C overnight. G1/S nucleosomes were centrifuged at 10,000× *g* at 4 °C.

MNase digestion of either G1/S or mitotic nucleosome solutions was performed for native ChIP. Rabbit anti-mH2A1 antibody (ab183041, Abcam, Cambridge, MA, USA) was incubated with Protein A/G magnetic beads (16-663, Millipore, Billerica, MA, USA) at 4 °C for 1 h with rotation, following the recommended procedure by the supplier. The beads were subsequently incubated overnight at 4 °C with constant rotation with the collected nucleosomes of the G1/S or mitotic preparations.

Prior to library preparation, to obtain the DNA, we input either nucleosomes or ChIP-enriched mH2A1 nucleosomes, which were treated overnight at 56 °C with 1% SDS and 200 μg/mL Proteinase K (Invitrogen, Carlsbad, CA, USA). Following a pre-incubation with 100 μg/mL RNase A (Invitrogen, Carlsbad, CA, USA) to eliminate RNA contamination, the released DNA was purified using phenol-chloroform and ethanol precipitation. Sequencing libraries were generated using approximately 1 ng of DNA per sample with the NEBNext^®^ Ultra™ II DNA Library Prep Kit (E7645S, NEB, Ipswich, MA, USA), and sequencing was conducted on the Illumina^®^ NovaSeq 500 or 6000 platform.

### 2.3. nChIP-Seq Data Processing

We utilized TrimGalore [17] to remove sequencing adapter sequences and low-quality bases (Q < 20) from raw sequencing reads. We used Bowtie2 [18] to align the trimmed single reads to human genome (GRCh38) and took Samtools [19] to remove the unknown chromosome segment mapped reads, unmappable reads, and multi-mapped reads. The mapped reads were converted to Bam format, then de-duplicated by Sambamba [20] markdup. We used bamCoverage to transform Bam files to BigWig files. The Pearson correlation between these was calculated using multiBamsummary with 2 kb bins and visualized with the plotCorrelation with the default parameters of Deeptools [21]. The mapped Bam files were used to identify mH2A1 enriched domains after normalization by the control (input nucleosome DNA by normR [22]). We divided the genome into 2 kb bins to count the total number midpoint of every mapped read in each bin. The significant mH2A1-enriched bins identified by Fisher’s test (*p* < 0.05) used exportR to output into Bed files and BigWig files. Adjacent bins were merged into mH2A1 domains for downstream domain-coverage analysis by BEDTools [23], and the coordinates of mH2A1 domains were displayed by Integrative Genomics Viewer [24].

### 2.4. RNA Data Analysis

Raw RNA sequencing data from unsynchronized ARPE-19 (GSM7998496) [25], HeLa cell (PRJNA30709) [26], and different cell cycle phases of ARPE-19 (GSE308366, SRP479011) [16], HeLa (GSE81485) [27] were downloaded for re-analysis. As described in the ChIP-seq data process, quality-filtered reads were aligned to the human genome (GRCh38), while uniquely mapped reads remained. Raw counts were generated by featureCounts [28], with unsynchronized samples used for FPKM (Fragments Per Kilobase of transcript per Million fragments mapped) calculation and synchronized samples for differential expression analysis by DESeq2 [29]; genes with an adjusted *p*-value (Padj) < 0.01 and an absolute log2 foldchange greater than 2 were considered significantly differentially expressed. RNA-seq coverage files were exported from Deeptools [21].

## 3. Results

### 3.1. Human Histone Variant mH2A1 Exhibits Widespread Genome Occupancy During Both G1/S and Mitotic Phases

To investigate the cell-cycle-dependent distribution of the histone variant mH2A1 in human cells, we collected G1/S and mitotically synchronized ARPE-19 and HeLa cells. We then performed nChIP-seq to isolate nucleosomes containing mH2A1. After single-end sequencing and alignment to the human reference genome (hg38), mH2A1-enriched regions were identified using the normR algorithm (*p* < 0.05) [22]. Based on the previous study in mice [9], we first examined the genome-wide dynamics of mH2A1 in ARPE-19 cells across the cell cycle. We found that mH2A1-associated domains occupied 64.3% of the genome during the G1/S phase and expanded to 69.6% during mitosis. In comparison, HeLa cells exhibited a higher proportion of mH2A1 enrichment, with 68.6% coverage in G1/S and 79.4% during mitosis. To characterize stable deposition patterns, we defined conserved regions as genomic loci occupied by mH2A1-containing nucleosomes in both G1/S and mitosis, indicating no deposition or removal events across the cell cycle. These conserved domains spanned 50.5% of the genome in ARPE-19 and 60.4% in HeLa cells. This extensive overlap suggested that mH2A1 played a critical role in maintaining epigenetic stability through mitosis. Further analysis revealed that in ARPE-19 cells, 53.1% of conserved mH2A1 domains overlapped gene bodies, while 46.9% were located in intergenic regions. In HeLa cells, the proportions were similar: 53.0% within gene bodies and 47.0% in intergenic regions.

Despite the distinct mH2A1 deposition patterns and broader overall domain coverage, the proportion of conserved domains occupying the genomic structure remained nearly identical between cell types. Specifically, we identified 117,847 and 79,096 mH2A1 domains in ARPE-19 and HeLa cells, respectively, during the G1/S phase, with a median size of approximately 4 kb. During mitosis, the number increased to 215,102 and 86,279 domains, respectively, with a larger median size of 6 kb.

Notably, although ARPE-19 and HeLa cells displayed comparable proportions of mH2A1-enriched regions at each cell cycle stage, their genome-wide distribution patterns differed considerably, with a Pearson correlation of 0.51 during G1/S and 0.56 during mitosis (Figure 1C). Nevertheless, their shared mH2A1-enriched regions span a substantial portion of the genome: 58.2% in G1/S and 59.6% in mitosis. To illustrate this, we examined chromosome 19 and observed that both ARPE-19 and HeLa cells exhibit similar local enrichment patterns of mH2A1 during G1/S and mitosis, highlighting areas of consistent deposition (Figure 1D).

### 3.2. Dynamic of mH2A1-Enriched Domains Are Conserved from Mouse to Human

The summary statistics of mH2A1-enriched genomic structure regions revealed that, during G1/S, domain coverage increased in intergenic regions and decreased slightly within gene bodies across all three cell types examined. Moreover, the proportional distribution across genomic annotations was comparable between G1/S and mitosis. Notably, the accumulation of mitotic mH2A1 enrichment was evident in each cell type, suggesting a conserved mechanism of domain dynamic across species (Figure 1A,B and Appendix A).

To assess the transcriptional impact of mitotic-specific mH2A1 domains, we analyzed all protein-coding genes using public RNA-seq datasets for ARPE-19 (GSM7998496) [25] and HeLa (PRJNA30709) [26]. We defined protein-coding genes with >80% coverage by mitotic-specific mH2A1 domains as *mH2A1-deposition-associated* genes (Figure 2A). In ARPE-19, this gene set (3737) had a median FPKM of 7.98, significantly higher (*p*-value *=* 6.61 × 10^−45^, Welch’s two-sample *t*-test) than genes not covered by these domains (median 0.39); 87.6% had FPKM > 1. HeLa showed a similar pattern: the mH2A1-deposition-associated genes (3958) had a median FPKM of 10.30 (over 90% of this gene set with FPKM > 1), markedly higher than uncovered genes (median 0.18). To assess the relevance between mH2A1 deposition and cell cycle-dependent transcription changes, we present the following analysis. In HeLa cells, only 13 genes were identified as differentially expressed genes (DEGs) between G1/S and mitotic phases, and among these, ~0.1% were mH2A1-deposition-associated genes (Figure 2B). By contrast, in ARPE-19 cells, approximately 12% of all expressed genes were cell cycle-dependent DEGs with 240 of them belonging to mH2A1-deposition-associated genes, corresponding to 6.4% of this gene set (Appendix A). The detailed cell-cycle-dependent DEG lists were provided in attached files (Appendix A). These findings revealed an unnoticed connection between mH2A1 deposition and cell-cycle-dependent transcriptional changes. Although ARPE-19 cells displayed thousands of cell-cycle-dependent DEGs, only a minor fraction corresponded to mH2A1-deposition-associated genes. These results indicate that actively expressed genes during G1/S are deposited by mitotic-specific mH2A1 domains, suggesting a role in transcriptional silencing during cell division.

Among the co-deposition-associated genes in HeLa and ARPE-19 cells, we observed similar patterns of mH2A1 deposition during mitosis. Representative examples included *GDAP2*, *WDR3* and *SMC1A*, which showed mH2A1 enrichment over their gene bodies exclusively during mitosis (Figure 2C). This mitotic-specific enrichment mirrors patterns previously observed in mouse 3T3 cells and our previous research results, which indicated that mH2A1 contributes to transcriptional silencing [6,7,9].

Interestingly, dynamic mH2A1 domains activity appeared more pronounced within intragenic regions in mouse 3T3 cells compared to humanARPE-19 cells. Homology analysis identified 1551 conserved genes between mice and humans (Appendix A), many of which were involved in essential cellular functions such as organelle biogenesis and chromatin organization. Meanwhile, the distribution of mH2A1 domains was also cell type- specific: 2105 genes uniquely regulated by mH2A1 in 3T3 cells were associated with embryonic development (Appendix A).

### 3.3. Dynamic mH2A1 Domains Are Implicated in Tumorigenesis

As mentioned above, HeLa cells also exhibited cell-cycle-dependent mH2A1 deposition; however, notable differences existed compared to non-malignant ARPE-19 cells. Specifically, HeLa showed a significantly reduced proportion of dynamic mH2A1 domains, alongside a marked increase in conserved domains across both cell cycle phases (Appendix A). These patterns suggest a stabilization or fixation of mH2A1 binding in malignant cells [30,31].

Moreover, during mitosis, HeLa cells demonstrated strong mH2A1 accumulation in genomic regions that display weak or no signal in ARPE-19 cells (Figure 1C). Among the mH2A1-deposition-associated genes in HeLa, we identified several highly expressed genes that, in contrast, showed minimal RNA expression and complete mH2A1 coverage in ARPE-19 across the cell cycle. Thus, these were not associated with mH2A1 deposition in ARPE-19 but are specifically regulated in HeLa and potentially linked to tumorigenesis (Figure 3A). For instance, *SLC38A2*, which encodes a glutamine transporter, plays a critical role in sustaining cancer cell proliferation, survival, and migration [32,33]. Likewise, *TBX3*, a member of the T-box family of transcription factors, functions as a direct repressor of p21 expression and has been implicated in a broad spectrum of cancers, including breast, ovarian, cervical, pancreatic, bladder, liver, and melanoma [34,35]. Our findings suggest that mH2A1 domains play a crucial role in the regulation of oncogenes in malignant cells. Although the exact mechanisms remain to be elucidated, this insight might inform future therapeutic strategies targeting epigenetic regulation in cancer.

We identified 1215 mH2A1-deposition-associated genes shared between HeLa and ARPE-19 cells (Appendix A). These genes were functionally enriched in essential mitotic processes, including chromatin remodeling and organelle biogenesis, as mentioned above (Figure 3B). In contrast, mH2A1-deposition-associated genes that were unique to HeLa during metaphase showed additional enrichment in cell-cycle-function-related pathways, such as DNA replication, DNA repair, and chromosome segregation [36,37,38,39,40] (Figure 3B).These findings suggested that in cancer cells, mH2A1 might reinforce chromatin stability in critical genomic regions, thereby supporting rapid proliferation and maintaining genomic integrity during cell division.

## 4. Discussion

In this study, we systematically analyzed the genome-wide distribution and cell-cycle-dependent dynamic mH2A1 domains in two human cell lines: ARPE-19 and HeLa. Both cell types exhibited a combination of phase-specific and conserved mH2A1 domains throughout the cell cycle. Notably, conserved domains occupied a significantly larger portion of the genome compared to dynamic ones. Although ARPE-19 and HeLa shared a relatively high degree of overlap in mH2A1-covered regions within the same cell cycle phase, their overall distribution patterns differed substantially. However, specific local chromosomal regions displayed similar enrichment patterns, suggesting partial conservation of domain placement. Additionally, the distribution of conserved mH2A1 domains between gene bodies and intergenic regions remained consistent across the two cell lines.

To assess cross-species conservation, we examined whether dynamic mH2A1 domains were similarly regulated in other organisms. We observed a consistent increase in mH2A1 occupancy within intergenic regions during G1/S, accompanied by a slight reduction within gene bodies [9]. Previously, Gamble et al. (2010) demonstrated that mH2A1 forms large conserved domains across different cell types [41]. Our results extended this concept, finding that dynamic mH2A1 domains in eukaryotic genome support an evolutionary conservation across species, as this pattern plays a critical role during mitosis in maintaining genome integrity and offspring phenotype. Specifically, in G1/S, mH2A1 domains tending to enrich in intergenic regions might contribute to the suppression of aberrant transcription, while in mitosis, dynamic mH2A1 domains covering intragenic regions could facilitate chromatin condensation and proper segregation.

To further investigate the functional implications of dynamic enrichment, we analyzed the protein-coding genes associated with mitotic-specific mH2A1 domains. The number of mH2A1-deposition-associated genes was comparable between ARPE-19 and HeLa cells, with the majority of these genes being highly expressed. Given that RNA polymerase II activity could disrupt nucleosome occupancy and alter chromatin organization, our findings aligned with previous observations in mouse 3T3 cells [9]. These results support the idea that mH2A1 domains are transiently deposited on active genes during mitosis, potentially contributing to temporary transcriptional silencing. Moreover, we further examined the potential regulation of mH2A1 deposition on *cis*-regulatory elements. Based on annotated enhancer sets [42], we integrated ATAC-seq data of ARPE-19 [43] and HeLa [44] cells to ensure enhancer elements were chromatin-accessible and cell-specific. By the above processing, we finally selected enhancers covered by mitotic-specific mH2A1 domains as *mH2A1-deposition-associated enhancers*, of which there were 11,937 in ARPE-19 cells compared to 22,137 in HeLa cells. This finding supports the perspective that enhancer reprogramming and aberrant activity contribute to the dysregulated transcriptional programs underlying tumorigenesis [45]. Based on published maps of enhancer–promoter contacts, we used ±125 kb of TSSs as the regulatory window [46], identifying 3881 enhancers in 88% of ARPE-19 mH2A1-associated-genes and 6957 in 97% of HeLa’s. These results suggest that mH2A1 depositions not only accompany direct intragenic coverage but may also exhibit cell-cycle-dependent regulation of enhancers. Collectively, this evidence reinforces the role of mH2A1 as a mitotic bookmark, helping stabilize chromosome states through cell division.

One of the most striking findings was the identification of a distinct subset of mH2A1-deposition-associated genes that were exclusive to HeLa cells. In ARPE-19, these gene bodies were consistently covered by mH2A1 throughout the cell cycle. In contrast, HeLa cells exhibited deposition, with mH2A1 enrichment restricted to mitosis and removed during the G1/S phase. These HeLa-specific mH2A1-deposition-associated genes were functionally enriched in fundamental cellular processes, including DNA replication, DNA repair, and chromosome segregation—key processes that support the rapid proliferation characteristic of cancer cells. In future work, we should precisely determine pathways specifically linked to tumorigenesis, thereby strengthening the biological relevance of our findings. Meanwhile, we also identified a group of shared mH2A1-deposition-associated genes that were highly expressed in ARPE-19 bet significantly repressed in HeLa. Examples include *MANF*, *DCBLD2,* and *PLK2*, which have been linked to tumor-suppressive functions in various cancers [47,48,49]. These contrasting patterns offer a potential explanation for the seemingly contradictory roles of mH2A1 reported in previous studies, where it has been described as both a tumor suppressor and an oncogenic factor. Our findings suggested that these opposing functions might coexist within the same malignant genome [50], indicating that mH2A1 is not inherently tumor-suppressive or oncogenic, but rather context-dependent.

In summary, our study provided an epigenetic framework for understanding the regulatory role of mH2A1 in cancer. Further investigation into isoform-specific functions of mH2A1.1 and mH2A1.2 across normal and malignant tissues will be essential for clarifying its paradoxical involvement in oncogenesis. Current evidence indicates that isoforms mH2A1.1 and mH2A1.2 exhibit transcriptional silencing concordance, while in cancer research, the ambiguous translational regulation of mH2A is cell- and cancer-type specific: in lung cancer, mH2A1.1 functions as an endogenous inhibitor of poly ADP-ribosyl polymerase I (PARP-1), linked to tumor-suppressive roles [13]. In HER2-positive breast cancer, HER2-induced expression of mH2A1.2 appears to feed back to enhance HER2 signaling, potentially contributing to increased tumor aggressiveness [51,52]. Thus, the balance between mH2A1 isoforms and their post-translational regulation likely underpins the context-dependent tumorigenesis of mH2A1, emphasizing the need to evaluate isoform dynamics when considering mH2A1 as a therapeutic target or biomarker.

## Figures and Tables

**Figure 1 biomolecules-15-01386-f001:**
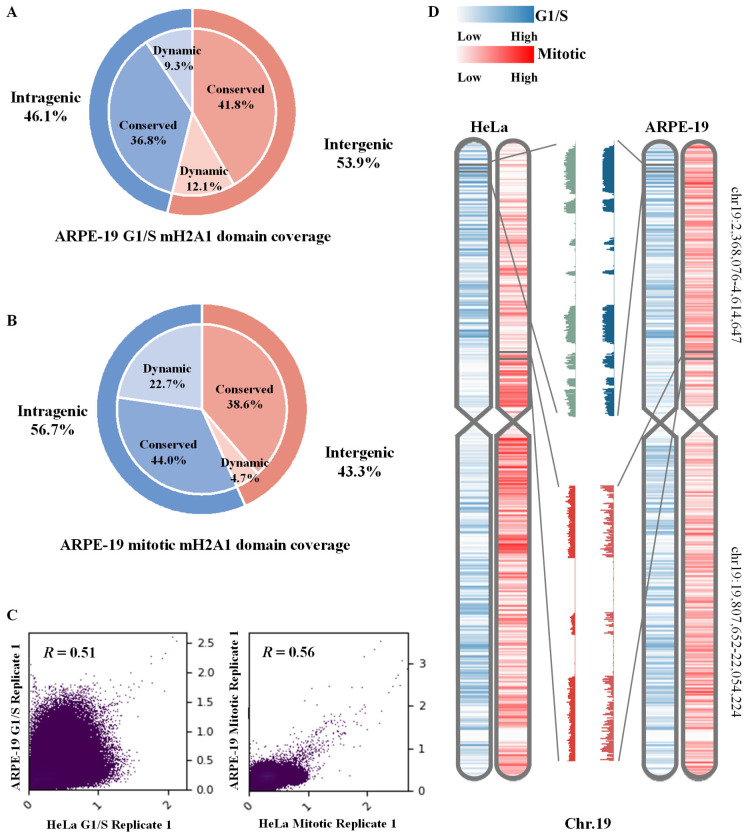
mH2A1 shows broad genomic coverage across different cell cycle stages in human cells. (**A**,**B**) depict the proportions of mH2A1 domain coverage across the genome during G1/S and mitotic phases, categorized into dynamic and conserved regions. (**C**) Genome-wide correlation of mH2A1 domain distribution between ARPE-19 and HeLa during the G1/S and mitotic phases. (**D**) presents a chromosome-wide heatmap of HeLa (**left**) and ARPE-19 (**right**) mH2A1 enrichment on Chr.19 during different cell cycles, with signal intensity indicated by color scale bar.

**Figure 2 biomolecules-15-01386-f002:**
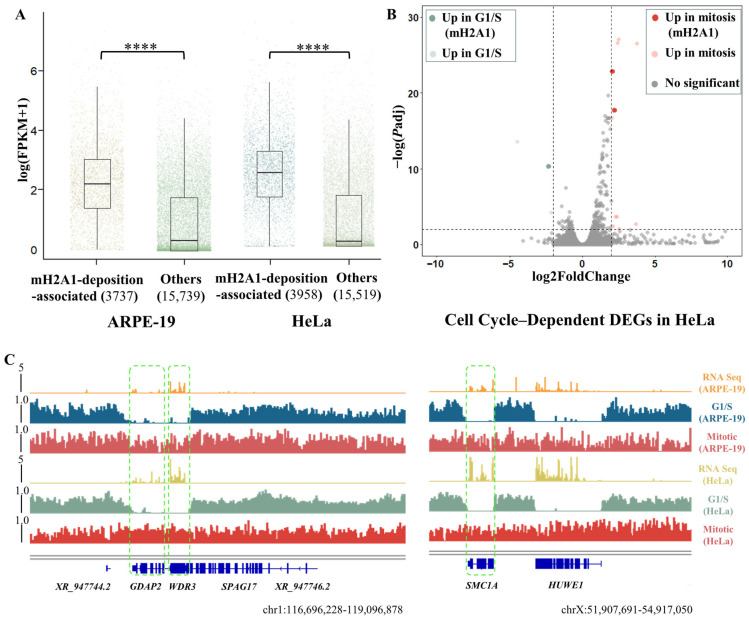
Expression patterns and co-location of mH2A1-deposition-associated genes across cell types. (**A**) Boxplot statistics of mH2A1-deposition-associated genes in ARPE-19 and HeLa cells, respectively. The horizontal line within each box represents the median. Welch’s two-sample *t*-test *p*-value: 6.61 × 10^−45^ (ARPE-19); 4.49 × 10^−56^ (HeLa). **** indicates *p*-value < 0.0001. (**B**) Cell-cycle-dependent differential expression analysis of HeLa cells. Genes upregulated in G1/S are shown in light reseda green, with mH2A1-deposition-associated genes highlighted in reseda green. Genes upregulated in mitosis are shown in light crimson, with mH2A1-deposition-associated genes highlighted in crimson. No significant genes are shown in gray. (**C**) Representative co-loci are shown on Chr.1 and Chr.X of ARPE-19 and HeLa cells; mH2A1-deposition-associated genes are highlighted with green dashed boxes. FPKM of co-deposition genes in ARPE-19 and HeLa cells: *GDAP2* (6.97; 7.79), *WDR3* (15.70; 18.96), *SMC1A* (11.76; 45.95). Tracks from top to bottom: ARPE-19 RNA-seq signal (orange), mH2A1 domains in ARPE-19 G1/S (navy), and in ARPE-19 mitosis (carmine), HeLa RNA-seq signal (yellow), mH2A1 domains in HeLa G1/S (reseda green), and in HeLa mitosis (crimson).

**Figure 3 biomolecules-15-01386-f003:**
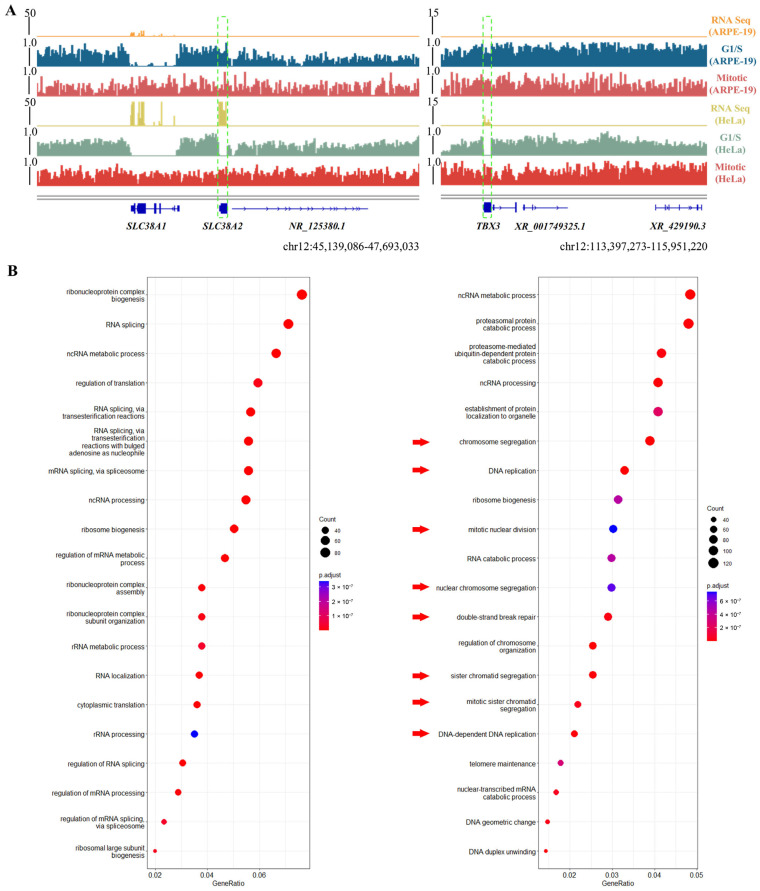
Oncogenesis-related and functional enrichment analysis of in HeLa and ARPE-19. (**A**) HeLa specific mH2A1-depostion-associated genes are oncogenesis-related and highlighted with green dashed boxes. FPKM of co-deposition genes in ARPE-19 and HeLa cells: *SLC38A2* (0.09; 366.36), *TBX3* (0.24; 37.72); (**B**) GO functional enrichment analysis. The left panel shows enriched terms for shared genes, while the right panel shows those for HeLa-specific genes. The red arrowhead marker items are commonly recognized as tumorigenesis related pathways.

## Data Availability

The datasets presented in this study can be found in the online repository (https://www.ncbi.nlm.nih.gov/geo/query/acc.cgi?acc=GSE303932, 20 September 2025) under the accession number GSE303932.

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
