# Peer review of "Regional Conservation and Transcriptional Regulation of Tumor-Associated Genes by macroH2A1 Deposition in Mammalian Cells"

_biomolecules, 2025, doi:10.3390/biom15101386_

Round 1
Reviewer 1 Report
Comments and Suggestions for Authors
The study by Deng et al. investigated the role of macroH2A1 deposition in regulating tumor-associated genes in human cell lines (ARPE-19 and HeLa) and mouse 3T3 cells, focusing on its dynamic behavior across the cell cycle and its implications for cancer biology. The use of native chromatin immunoprecipitation followed by sequencing (nChIP-seq) to explore macroH2A1 distribution is a robust approach, and the findings regarding conserved and cancer-specific patterns are intriguing. The identification of macroH2A1’s role in transcriptional regulation, particularly its enrichment in actively expressed genes during mitosis, adds valuable insights to the field of epigenetic regulation in cancer. However, the manuscript has some limitations. Below are specific theoretical and experimental comments for the authors to address.
Theoretical comments:
- The manuscript notes that macroH2A1 can act as both a tumor suppressor and an oncogenic factor (Page 10), which is consistent with prior literature (e.g., Novikov et al., 2011; Dardenne et al., 2012). However, the discussion lacks a deeper exploration of the molecular mechanisms underlying this dual functionality. For instance, how do specific isoforms (macroH2A1.1 vs. macroH2A1.2) or post-translational modifications influence its context-dependent role? The authors should elaborate on these mechanisms, potentially referencing studies on isoform-specific functions in cancer, from Buschbeck's or Vinciguerra's labs, for instance.
- The study highlights conserved macroH2A1 domains across human and mouse cells (Page 5, Figure 5A). However, the theoretical basis for this conservation is not sufficiently discussed. What evolutionary pressures might drive the conservation of macroH2A1 deposition patterns, particularly in intergenic regions? The authors should provide a hypothesis or cite relevant literature (e.g., Gamble et al., 2010) to frame the significance of this finding.
- The manuscript identifies a cancer-specific pattern of macroH2A1 distribution in HeLa cells associated with pro-oncogenic genes (Page 2). How might macroH2A1’s enrichment in DNA replication and repair genes (Page 9) contribute to the hallmarks of cancer, such as genomic instability or sustained proliferative signaling? The authors should connect their findings to established cancer biology frameworks.
- The study reports that macroH2A1 is enriched at actively expressed genes during mitosis (Page 5), which contrasts with prior findings that it contributes to transcriptional silencing (Page 5, citing previous research). This apparent contradiction is not adequately addressed.
Experimental comments
- The Materials and Methods section (Page 2) mentions cell culture and synchronization but lacks details on critical experimental parameters. For instance, how was macroH2A1 enrichment quantified, and what controls were used to validate specificity? The authors should provide a detailed protocol, including references to established methods (e.g., Langmead et al., 2009, for sequence alignment, cited on Page 11).
- The statistical significance reported for FPKM differences in ARPE-19 and HeLa cells (Page 5) is convincing, but the methodology behind these p-values is unclear. Were multiple testing corrections applied? What statistical tests were used (e.g., Wilcoxon rank-sum test)?
- The study uses ARPE-19 (normal retinal epithelial cells) and HeLa (cervical cancer cells) but does not justify this choice beyond their synchronization properties. Why were these specific cell lines selected, and how do they represent broader normal and cancerous cell types? The authors should discuss the relevance of these models.
- The observation of increased macroH2A1 domain coverage during mitosis (Page 4) is interesting, but it is unclear whether this is specific to macroH2A1 or a general feature of histone variants. Did the authors perform parallel ChIP-seq for other histones (e.g., H3 or H2A.Z) as controls? Including such controls would strengthen the claim of macroH2A1-specific dynamics.
- The mention of SPEC, CLA, and TCDS as representative genes with mitotic macroH2A1 enrichment (Page 5) lacks context. Are these genes known to be involved in cancer or cell cycle regulation? The authors should provide background on these genes and validate their findings with orthogonal methods (e.g., qPCR or Western blot) to confirm macroH2A1’s role in their regulation.
Author Response
Please refer to the attached PDF file.

Reviewer 2 Report
Comments and Suggestions for Authors
The authors of the manuscript Yongzhuo Deng et al have presented in their manuscript entitled "Regional Conservation and Transcriptional Regulation of Tumor-Associated Genes by macroH2A1 Deposition in mammalian cells" new epigenetic insights into the role of macroH2A1 in malignant proliferation. The authors show macroH2A1’s genome-wide distribution and dynamic regulation are context-dependent and show a stark difference between normal and cancer cells. It may act as both tumor suppressor and oncogene depending on its regulation, suggesting the need to study its isoforms (macroH2A1.1, macroH2A1.2) and could pave way for future cancer therapeutic platforms. The authors have done a commendable job in presenting their cross species comparison and highlighting the dual roles of MacroH2A loading and a context dependent role of MacroH2A. Overall the manuscript can be considered for publication.
Author Response
Comments: The authors of the manuscript Yongzhuo Deng et al have presented in their manuscript entitled "Regional Conservation and Transcriptional Regulation of Tumor-Associated Genes by macroH2A1 Deposition in mammalian cells" new epigenetic insights into the role of macroH2A1 in malignant proliferation. The authors show macroH2A1’s genome-wide distribution and dynamic regulation are context-dependent and show a stark difference between normal and cancer cells. It may act as both tumor suppressor and oncogene depending on its regulation, suggesting the need to study its isoforms (macroH2A1.1, macroH2A1.2) and could pave way for future cancer therapeutic platforms. The authors have done a commendable job in presenting their cross species comparison and highlighting the dual roles of MacroH2A loading and a context dependent role of MacroH2A. Overall the manuscript can be considered for publication.
Response: We sincerely thank the reviewer for the positive evaluation and for recognizing the significance of our findings. We fully agree with the assessment that macroH2A1 plays a context-dependent role in cancer biology, acting as both a tumor suppressor and oncogene depending on isoform regulation. We are encouraged by your acknowledgement of our cross-species comparative approach and our efforts to highlight the regulation of macroH2A1 deposition. This valuable feedback reinforces our view that further isoform-specific investigations will be critical for understanding macroH2A1’s potential as a therapeutic target.
Reviewer 3 Report
Comments and Suggestions for Authors
General Feedback:
Deng et. al, sought to understand the dynamics of macroH2A domains in the context of the cell cycle state, furthering knowledge on the macro-histone and its contributions to gene expression. The manuscript presents two central claims: (1) that macroH2A deposition occurs at a specific time in the cell cycle with implications in expression, and (2) that macroH2A isoforms exhibit differential roles in cancer, with one more closely associated with tumor-suppressive activity and the other with oncogenic activity. These points are already documented in the literature: Cell cycle–specific deposition of macroH2A, particularly retargeting and incorporation during G1 after mitosis, has been reported previously (e.g., Chadwick & Willard, 2002; Sato et al., 2019). Isoform-specific roles in cancer biology have been described in multiple contexts, with macroH2A1.1 often functioning in a tumor-suppressive manner and macroH2A1.2 displaying context-dependent pro-oncogenic properties (e.g., Re et al., 2016; Hsu & Shia, 2021; Recoules et al., 2022). However, the consequences in the expression patterns during cell cycle are not well defined. After arresting cells in G1/S phase, they performed native ChIP-seq on ARPE-19 and HeLa during different stages of the cell cycle and then compared with expression patterns for these cells using publicly available data. However, this manuscript is missing some key supporting data to define the connections between macroH2A occupancy and gene expression.
Major Comments:
In order to establish direct links between cell cycle deposition and expression the authors should perform RNA-seq analysis in G1/S phase and Mitosis in both cell lines to have a direct comparison and to see if the distribution of domains is indeed following the patterns of expression. This will also allow for a more in depth analysis of the differences between the two cell lines. The expression analysis is currently very shallow.
It is unfortunate that the authors chose two cell lines that have not been previously mapped for macroH2A enrichment so we do not have a clear sense of what kind of information we are gaining from arresting cells and profiling them at different time points. It would be good to have the nChIP done in asynchronous cells as a base line to compare too.
MacroH2A analysis: the authors used a not so standard method for defining macroH2A domains. They should at least compare their method with known methods of peak calling such as MACS2 and/or SICER that have been developed for large domains.
Finally, the authors need to show data to verify the G1/S and Mitotic states, either by immunofluorescent cell cycle staining or flow cytometry. Instead the authors only reference previous papers published from the lab.
Minor comments:
While it has been already mentioned that the two cell lines used have not been previously profiled the authors should at least explain why these cell lines were specifically chosen in the context of the oncogenic claims that they are making.
The authors could also look at the differences in deposition at enhancer elements given the role of macroH2A in enhancer regulation, and whether those are affected by cell cycle.
Author Response
Please refer to the attached PDF file.

Round 2
Reviewer 1 Report
Comments and Suggestions for Authors
All comments have been addressed.
Author Response
We sincerely thank you once again for your thoughtful and constructive comments in the previous round. Your suggestions guided us to provide more detailed experimental information, including the validation of cell synchronization accuracy and antibody specificity used in ChIP. You also precisely pointed out omitted terms and references, moreover, allowed us to refine the structure of our results and disscusion. Finally, your feedback made us realize the insufficiency of our background research, particularly in the selection of cell lines and prompted us to strengthen the discussion by incorporating a more comprehensive perspective on the role of macroH2A1 isoforms in cancer research. We are truly grateful for your insightful guidance, which has greatly improved the clarity and overall impact of our manuscript.
Reviewer 3 Report
Comments and Suggestions for Authors
Unfortunately, the authors did not address the major comments experimentally and argued about the relevance of such suggestions. We believe that is there is any relevance to the analysis of the chromatin changes without looking at the most direct outcome (transcription) the premise of the paper falls in void. The authors mention that "We fully agree that RNA-seq analysis of synchronized cells in G1/S phase and mitosis would indeed provide more direct evidence of macroH2A1 deposition influences transcriptional activity. In our current study, we rely on published RNA-seq datasets from unsynchronized cells. As the untreated cells in normal culture are in G1/S, representing gene expression under physiological conditions. Moreover, during mitosis, transcription is largely silenced due to the highly condensed chromosomes, and published results and our previous work have demonstrated that the majority of RNA extracted from metaphase chromosomes corresponds to structural RNA preserved from preceding interphase, rather than nascent transcripts." We transcription changes are mostly silenced we would argue that the chromatin analysis in different cell cycle states would be irrelevant. The differences that may be detected in expression may be critical to understand the differences in chromatin.
The enhancer analysis is also really unsatisfactory. The authors only mention the number of peaks associated with ATAC-seq but do not discuss the differences between cell cycle states and their correlation with transcription.
Round 3
Reviewer 3 Report
Comments and Suggestions for Authors
The authors have now been able to include RNA-seq datasets to try to address the major criticism of the reviewer and the central hypothesis of this paper. We still believe that if is there is any relevance to the analysis of the chromatin changes without looking at the most direct outcome (transcription) the premise of the paper falls in void. Without repeating our previous comments our major concern remains unsolved with the presented response to the reviewer. Since this is central to the paper, we believe the RNA-seq data should be published and analyzed accordingly, and not shown as a response to the reviewer. The simple Welch’s two-sample t-test analysis of a subset of genes pooled together does not exclude differences in specific genes. The data should be included in the paper as a figure and global differential gene expression should be performed and discussed in light of the mH2A domains.
For the enhancer analysis, albeit very superficial, is is satisfactory since it is not central to the paper’s findings.
Round 4
Reviewer 3 Report
Comments and Suggestions for Authors
We thank the authors for the prompt response and understand the eagerness to publish their manuscript. However, since the expression data is available the request was not purely to state the absence of significant differences or quantify those by percentage of genes that may be changing but to describe exactly the RNA-seq analysis. We believe it is critical to clarify the RNA-seq findings and maybe it will be necessary in this case to be very specific to prevent further delays in the process. The revision should include the following 1) Graph (not table) of RNA-seq differential expression analysis for both cell types (eg. volcano plots). Highlighted in those volcano plots should be the significantly different genes (using a threshold) and the genes that have macroH2A (with a different color or in a separate plot). If space is an issue, Figure 3 B and C can certainly be consolidated (there is currently too much empty space for little information. 2) Table of genes that are both significantly different and have macroH2A and genes that do not have macroH2A domains. 3) The data should also be fully accessible for public use if that is a requirement from the journal, and a link to the publicly available datasets should be included.
Round 5
Reviewer 3 Report
Comments and Suggestions for Authors
Thank you for addressing our comments and concerns.